# The Contributions of Food Groups to the Daily Caloric Intake in Mongolian Population: A Mon-Timeline Study

**DOI:** 10.3390/nu13114062

**Published:** 2021-11-13

**Authors:** Oyuntugs Byambasukh, Anar Bayarmunkh, Agiimaa Byambaa, Anujin Tuvshinjargal, Delgermaa Bor, Urangoo Ganbaatar, Byambasuren Dagvajantsan, Tsolmon Jadamba

**Affiliations:** 1Department of Endocrinology, School of Medicine, Mongolian National University of Medical Sciences, Ulaanbaatar 14210, Mongolia; anarbayrmonh2406@gmail.com (A.B.); agiimaa0119@gmail.com (A.B.); anujintuvshinjargal1@gmail.com (A.T.); bdelgermaa77@gmail.com (D.B.); urangoo9918@gmail.com (U.G.); 2Department of Neurology, School of Medicine, Mongolian National University of Medical Sciences, Ulaanbaatar 14210, Mongolia; byambasuren@mnums.edu.mn; 3TimeLine Research Center, Ayud Tower, Ulaanbaatar 14240, Mongolia; 4Brain and Mind Institute, Mongolian Academy of Sciences, Ulaanbaatar 14200, Mongolia

**Keywords:** diet intake, food groups, lifestyle, daily caloric intake

## Abstract

(1) Background: The “Ger Recommendations” have been advised to promote a healthy diet in Mongolia. These recommendations emphasize the ratio of six macro-food components to ensure proper nutrition. In this study, the ratio of these six groups to the total daily caloric intake was determined. (2) Methods: This study was conducted as part of a study at the Clinical Cohort (“Mon-Timeline”) of the Mongolian National University of Medical Science. A macro-community ratio was calculated using a 24-h dietary recall diary of a total of 498 people. (3) Results: The mean age of the study participants was 43.9 ± 12.9 years. Among them, 21.8% (*n* = 110) were male. Of the total calories, 44.7% were grains, 29.2% were meat and protein products, 9.3% were fats, 7.1% were dairy products, 6.6% were vegetables, and 3.1% were fruits. According to the ratio of the six groups in the Ger Recommendations, meat and grains exceeded the recommended amount, while fruits, milk, and vegetables were consumed less. It has been observed that the older a person ages, the closer they are to following these recommendations. In terms of gender, women consumed more fruit and milk than men. (4) Conclusions: The ratio of macronutrients in the daily caloric intake of Mongolians is inadequate. Therefore, knowledge about the “Ger Recommendations” needs to be studied in relation to people’s healthy eating knowledge and attitudes. If necessary, the appropriate awareness needs to be increased to educate the public on proper eating habits.

## 1. Introduction

Mongolia is a landlocked country in Central Asia, located between China and Russia. For centuries, the Mongolian way of life has been nomadic, but since the 1990s, due to social changes, Mongolians have undergone significant changes in their lifestyle and eating culture and have become more sedentary [1,2]. Due to this and other lifestyle changes, the prevalence of many lifestyle disorders has increased rapidly in recent years. For instance, the prevalence of obesity increased from 17% in 1993 to 64.5% in 2019 [3,4]. In 1999, 3.2% of the population had diabetes mellitus type 2, but in 2019 it increased to 8.5% [3,5]. According to the World Health Organization, 64.1% of all deaths in Mongolia in 2015 were due to non-communicable diseases (NCDs) and chronic diseases, which is 6% less than the number of NCD-related deaths worldwide [6]. The WHO also notes that one in three Mongolians dies of diabetes, cardiovascular disease (CVD), and cancer before the age of 70 [6]. In 2017, one in five deaths in Mongolia was due to cancer, and 75 percent of them were related to lifestyle, habits, and preventable cancers of the liver, stomach, and cervix, according to the Ministry of Health and the World Health Organization [6].

One of the main risk factors for NCDs mentioned above is unhealthy diet. In order to promote healthy diet in Mongolia, the “Ger Recommendations” are advised in a way that is understandable to its population [7]. The “Ger” is a traditional Mongolian felt house in which Mongolians have lived for centuries. It is not only a traditional dwelling, but also a unique intangible cultural heritage that conveys the creative, unique thinking and skills of nomadic people by incorporating many things such as handicrafts, sewing, felt handicrafts, construction techniques, patterns, and symbols of interior design [8] (Figure 1). Mongolians have been aware of the “Ger Recommendations” for at least 20 years. Recently, they have been reflected in the order of the Minister of Health in line with the Mongolian body’s recommendations on the sources of nutrients that must be obtained from food on a daily basis [7]. these recommendations outline the basic principles of a healthy diet, such as eating a variety of foods, eating three or more servings of vegetables, and eating at least two servings of fruits a day [7]. However, no study has yet been conducted on how this recommendation is followed among Mongolians [9,10,11].

“Mon-Timeline” is the name of the multidisciplinary, population-based, prospective cohort study conducted in Mongolia to investigate various health problems among Mongolians, especially those associated with oral, psychological, mental and neurocognitive health problems in 2020. This study established a national database based on data collected from Ulaanbaatar and four rural regions in Mongolia. In this study, we examined the characteristics of the Mongolian diet, especially the “Ger Recommendations”, as a percentage of the total daily caloric intake. We also aimed to differentiate between nutrition, gender, and age, and urban and rural disparities.

## 2. Materials and Methods

### 2.1. Data Source

This research was conducted as part of a study at the Clinical Cohort (“Mon-Timeline”) of the Mongolian National University of Medical Science. In the “Mon-Timeline” cohort study, the sample size was calculated based on the total populations aged 13 and 65 years (*n* = 2,084,034) and the latest prevalence of dental caries (*p* = 83%) assuming a 95% confidence interval (*Z* = 1.96) with a 2% acceptable margin of error (*e* = 0.02), which gave a sample size of 2709 persons. Study clusters (*n* = 64) and participants were randomly selected from 8 provinces in the Western, Mountain, Eastern, and Central regions according to geographical zoning and 6 districts of Ulaanbaatar city (Figure 2). The data were collected from the end of June to November 2020 with the approval of the medical ethical committee of the Mongolian National University of Medical Sciences ( METc: 2020/3-05).

During the data collection as mentioned above, there were multidisciplinary teams working at the same time, and to save time we were able to interview 40–50% of the participants in each cluster with 24-h dietary recall. A total of 1600 participants were interviewed with 24-h dietary recall. We included people aged 18 years and over (*n* = 1209) in this study. The exclusion criterion was missing data of 24-h dietary recall (*n* = 274). In addition, people who did not complete the form were excluded (*n* = 354). Furthermore, participants with a history of following a special diet due to any reason were excluded (*n* = 83). Finally, a total of 498 participants were included in the current analyses.

### 2.2. 24-h Dietary Recall

The survey was conducted utilizing a 24-h dietary recall, which consisted of 4 columns and 4 rows of a sample table that could record the quantity, method of preparation, amount of food, and where the person ate the food the day before or between meals. Before taking a 24-h dietary recall from the study participants, the table was carefully explained, instruction was given, and supplementary questions were answered. To make the participants comfortable and able to freely write, a desk and chair were placed in the questionnaire room, and they were asked to fill in the survey notes completely, accurately, and correctly. A 24-h dietary recall uses an organized interview and written method that provides detailed information about all foods, water, beverages, supplements, and snacks consumed by the respondent within the past 24 h, usually from the morning of the previous day. In several studies, a 24-h dietary recall has been described as the simplest, easiest, and most cost-effective way to assess people’s nutritional status. It has also been identified as one of the most suitable methods for use in low-income and low-literacy populations [12,13].

To calculate the results of the study utilizing this 24-h dietary recall, we initially recorded the names of each product on each person’s own page, along with the size, and divided them into appropriate food groups (cereals, fruits, meat and meat products, dairy products, fats, and vegetables). Then, according to the Japan Diabetes Society’s Food Exchange List, the appropriate amount of kcal was calculated based on the gram and size of each food item, from which a unit calculation was performed [14]. The unit size was determined by the following formula [14].

1 unit = 80 kcal

Unit size = Calorie intake of each food group/80 kcal.

### 2.3. “Ger Recommendations”

These recommendations were issued by the Ministry of Health of Mongolia to allow Mongolians to use the nutrients, food, and nutrition required for the six main types of food to ensure the proper functioning of the body in accordance with age, gender, living conditions, and employment [7]. Along with all the principles of healthy eating, this recommendation is divided into six groups: cereals, meat products, dairy products, vegetables, fruits, and fats. The “Ger Recommendations” for children, expectant and lactating women, adolescents, and adults (low-, medium-, and high-intensity work) are calculated as the aggregate number of units per day. In our study, we selected and implemented healthy eating recommendations for adults. Depending on the sex of adults with moderate physical activity, these recommendations include 9 to 10 units of cereals, 4 to 5 units of meat and meat products, 3 to 4 units of dairy products, 5 units of vegetables, 5 units of fruits, and 2 units of fat. Salt and sugar intake should not exceed 5 g per day, alcohol should be limited, and sugar intake should be less than 5%. The results were calculated by comparing the amount of food groups consumed per day by the respondents in accordance with these recommendations.

### 2.4. Other Variables and Measurements

The interview included questions related to participants’ demographic and lifestyle characteristics. Education and family income variables were categorized as low, medium, or high levels. Marital status was classified as a dichotomous variable, including married or cohabiting or single. Lifestyle characteristics including smoking, alcohol use, and physical activity were classified as dichotomous variables: smokers/non-smokers, never/use of alcohol, and physically active/inactive. Furthermore, participants’ body weight (in kg), height (in cm) and waist circumference were measured by well-trained assistants implementing a standardized protocol, and Body Mass Index (BMI; kg/m^2^) was subsequently calculated.

### 2.5. Statistical Analysis

General characteristics of the study population were expressed as means with a standard deviation (SD) and as numbers with percentages. The differences among groups were compared using the Student’s *t*-test and Pearson’s Chi-square test. The ratio of the six food groups in the daily caloric intake was calculated as a percentage for each food group.

To examine whether the food was consumed according to the “Ger Recommendations” for Healthy Eating in Mongolia, the amount of food consumed by relatively healthy people aged 18 and over was compared with the recommendations. In this additional analysis, we assumed that the sum of the total units of the 6 food groups in the “Ger recommendation” was 100 percent, and calculated the percentage of each group, and examined whether more or less than this amount was consumed. It was calculated as follows: 45.2%, 6.5%, 19.4%, 12.9%, 6.5% and 9.7% for cereals, fruits, meat, dairy products, fat, and vegetables, respectively.

For all the statistical analyses we used IBM SPSS V.27.0 (IBM, Chicago, IL, USA) and GraphPad Prism V.4.03 (GraphPad Software, La Jolla, CA, USA). A statistical significance level was set at *p* < 0.05 for all the tests.

## 3. Results

A total of 498 participants were included with a mean age of 43.1 ± 13.1 years. Among them, 21.8% (*n* = 110) were men. Some demographic and socioeconomic characteristics of the participants are shown in Table 1. Fewer women were smokers or consumed alcohol, and women were less physically inactive than men (*p* < 0.05).

Considering the ratio of the 6 food groups in the total daily caloric intake, grains (44.7%), meat and protein products (29.2%), and fat (9.3%) accounted for the majority, while dairy products (7.1%), vegetables (6.6%), and fruits (3.0%) accounted for a small percentage. In terms of gender, this ratio was relatively close, but the consumption of fruits and milk products was relatively low for men (Table 2). There were no gender differences in the six groups of food compared to the rural population, while the urban population had a statistically significant difference in fruit consumption between men and women. Furthermore, urban people consumed more fruit (4.8%) than rural people (2.1%), but fat consumption (11.0%) was higher than in rural areas (8.6%). In particular, urban women and men consumed significantly more fruits and fats compared to rural women and men, respectively (Table 2). Moreover, in rural areas, grain consumption was higher than in urban areas, especially in women.

In terms of age, the consumption of fruits and dairy products has been increasing with age. In addition, young women aged 18–35 years consume twice as many fruits and vegetables as men. The daily caloric intake of fruits and vegetables was 3.8% and 5.8% for women and 1.9% and 3.2% for men (Figure 3).

Fruit consumption has been observed to increase with age in men. For instance, it was 1.9% in the 18–35 age group, 3.6% in the 35–55 age group, and 3.9% over the age of 55. For the other groups, no significant differences in age or sex were observed.

In addition to the percentage ratio of the six food groups to the total daily caloric intake to find out how the six ratios of healthy eating in Mongolia are being followed, the respondents found that their consumption of grains (especially women), meat (especially men), and fats exceeded the recommended levels. The consumption of fruits, vegetables and dairy products was low (Figure 4). For those living in urban and rural areas, grains were consumed more than the recommended amount by mostly rural women and predominantly urban men. However, both urban and rural people consumed fewer fruits, vegetables, and dairy products than recommended in the “Ger Recommendations”.

## 4. Discussion

In this study, the ratio of the 6 groups of food to the total calories of the day was calculated as 100% of the total calories, and the percentage of the 6 groups and desserts was calculated as a percentage. In addition, according to the Ger Recommendations, the sum of the numerical values to be used by each of the 6 groups for adult men and women was found to be 100%, including the percentage of each of the six groups, and the percentage of over- and under-use. Thus, the ratio of total calories seems to be relatively maintained, but the consumption of fruits, vegetables, and dairy products in the Ger Recommendations is not as high as recommended. In other words, according to our research, it is not enough for Mongolians to follow the “Ger Recommendations”. The most common foods consumed by Mongolians are meat products, grains, and fats. It is noteworthy that despite the development of animal husbandry, Mongolians’ consumption of dairy products does not reach the level of the “Ger Recommendations”. The consumption of fruits and vegetables is also insufficient.

As mentioned earlier, Mongolia has a relatively small arable land area compared to other countries due to its traditional nomadic lifestyle and climatic factors (0.003%) [15]. As a result of global climate change, Mongolia’s average air temperature has risen by 2.07 degrees over the past 70 years, 0.85 degrees more than the world average in the past century. Studies have shown that in countries, the differences in diets are more dependent on their agricultural lands [15]. Thus, a small arable land area might be a reason for lower consumption of vegetables and fruits among Mongolians. Although agriculture has been developed in Mongolia recently, cereals still account for 75.6% of total agriculture [16]. According to statistics, by 2020 potato cultivations in Mongolia were predicted to provide 100% of the country’s supply, while vegetables and fruits were predicted to provide only 60.7% and 3.5%, respectively. As for fruits, the statistics mentioned above show that we import more than 90% of our fruit supply, which is the reason for the high price of fruits and berries. In addition, 92% of the fruits grown in Mongolia are sea buckthorn, and the remaining 8% are apples, cherries, currants, blueberries, and plums [16]. The appearance of apples, pears, and bananas was selective among the respondents, indicating that there was little public awareness of the fruits grown in Mongolia, that imported fruits were used for consumption, and that poor quality or cost due to transportation and storage of domestically produced fruits may be the reason for the low consumption [16,17]. The low consumption of vegetables in urban areas is likely to be due to seasonal crops and rising prices for imported vegetables, while in rural areas, especially in suburban soums, food safety and road transport issues are related to the above-mentioned reasons [17]. The group of foods that Mongolians consume the most are meat and fat. Because of the traditional nomadic lifestyle, we expected that rural people would consume more meat, but it was higher in urban men and women. Moreover, the dairy product consumption has not reached the recommended level in rural people. Thus, this is not so much a result of traditional nomadic lifestyles, but rather can be explained as a result of urbanization. Furthermore, grain consumption in urban areas is higher than in rural areas. The above-mentioned inadequacy of the proper proportions of food ingredients in urban and rural people might equally be related to people’s knowledge and attitudes. Therefore, there is a need to further study the knowledge, eating habits, and attitudes of Mongolians and, if necessary, the appropriate awareness needs to be increased in order to educate the public on proper eating habits.

In Mongolia, NCDs have accounted for about 60–80% of all deaths for the past 10 years, the most common of which are CVD, stroke, and lung cancer [4]. Another study found that 1 in 5 Mongolians aged 15–64 is more likely to develop a non-communicable disease [18]. The prevalence of chronic diseases among Mongolians is high and might be due to the excessive consumption of animal fats and the lack of sources and consumption of fruits and vegetables. A few previous studies found that one of the largest contributing factors to these disorders is an unhealthy diet [9,19]. According to a survey conducted by the National Statistics Office, the national average caloric intake and some basic nutrients per capita exceeded the recommendation [19]. The daily caloric intake of Mongolian adults is 2525 kcal, and 33.7% is from fat intake, which is 1.3 times higher than Japanese consumption [9]. Comparing the total daily caloric intake of Palau, Mongolia, and Japan, the caloric intake of Mongolians and Palau is similar to that of the Japanese [20]. In accordance with our study, a previous study found higher consumption of meat among Mongolians. Mongolia’s per capita consumption of meat products was 140 kg, which is much higher than the average Asian standard and that of developed countries [15]. Although Uruguay is a landlocked country, it is like Mongolia in terms of meat and animal fats, food, and unhealthy diets, with high rates of obesity in both men and women, and it has the 31st highest incidence of cancer in the world. Several studies in this country continue to suggest that poor diet may be a primary cause of many cancers [21,22]. A study assessing changes in the diet of the Russian population from 1990 to 2018 showed that consumption of meat and vegetables increased by 20–60% compared to the 1990s [23]. Compared to the 1990s, the consumption of fats has increased, and the consumption of cereals has decreased to some extent. However, despite the increase in vegetable consumption, it did not meet the WHO’s recommendation, and despite the decline in grain consumption, it exceeded the recommended level [23]. Several such studies have shown that the nutritional status of the Russian population is like that of Mongolia, with excessive consumption of unsaturated fats, salts, and sugars, as well as insufficient consumption of fruits and vegetables and a high risk of NCDs [24,25]. Furthermore, in 2019, the European Journal of Data Analysis published an article on the extremely low consumption of some European fruits and vegetables, in which Hungary ranked lower than any other European country [26]. It was also the fourth most obese country in the world in the same year, with one in five adults being obese [27,28]. Lifestyle diseases and NCDs accounted for the majority of deaths in Hungary. Eating habits and dietary risk factors alone account for 28% of all deaths. Hungary is a less active country in terms of meat and fat consumption compared to Mongolia, but on the one hand, the consumption of fruits and vegetables is very low, and on the other hand, obesity is due to excessive consumption of sugar and salt and poor diet. Complicated CVD, diabetes, and stroke in particular are among the most fatal diseases in the country [28]. Comparing several countries in this way shows the importance of diets, and how the introduction of proper nutrition and nutritional needs can reduce NCDs. In addition, our findings suggest that eating more meat and fewer fruits and vegetables may increase the risk of NCDs among Mongolians. Therefore, dietary recommendations need to be further improved.

Every country in the world has its own dietary guidelines for its people, depending on their customs, traditions, and climate. For example, countries such as the United Kingdom and Australia use a proper plate design, while the United States, France, and Finland exploit a food pyramid, and Canada uses a rainbow design [29]. For landlocked countries, the dietary guidelines recommend a balanced intake of seafood and a consumption of 500 g or more of vegetables. In other countries, moreover, they have adapted their diets and essential foodstuffs in an understandable way to their people. The difference may be that these recommendations are not only approved, but also promoted at both the educational and social levels, and in some countries, brochures are published for citizens. Mongolia, however, has not developed a long-established recommendation on proper nutrition, and in recent years has advised following the “Ger Recommendations”. Our study is insufficient to examine whether the “Ger Recommendations” are appropriate for Mongolians or whether more detailed recommendations are needed. Therefore, studies are needed to evaluate the knowledge of the “Ger Recommendations” among Mongolians in order to assess people’s knowledge and attitudes towards healthy eating. If necessary, the appropriate awareness needs to be increased to educate the public on proper eating habits. Furthermore, it needs to increase and generate demand for nutritious food and preserve the culture and traditions of Mongolian cuisines.

The strength of this study, which examined the ratio of macro-nutrients in the daily caloric intake of Mongolians, sought to involve people from all over Mongolia, both urban and rural. In addition, the 24-h dietary recall diary method we used in this study may be more appropriate than other methods, such as asking for each food that may be included in the food groups listed in the “Ger Recommendations”. The limitation of the study is that a large number of participants were excluded due to the 24-h dietary recall diary not being adequate. Another limitation is that the number of male participants was small in this study.

## 5. Conclusions

The ratio of macronutrients in the daily caloric intake shows that common foods consumed by Mongolians are meat products and fats. The consumption of fruits, vegetables, and dairy products in the “Ger Recommendations” is not as sufficient as recommended.

## Figures and Tables

**Figure 1 nutrients-13-04062-f001:**
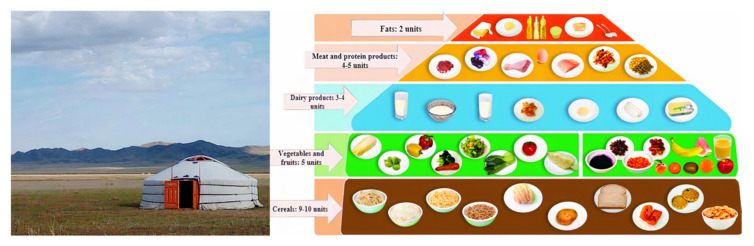
Mongolian nomadic felt house (Ger) and the “Ger Recommendations”.

**Figure 2 nutrients-13-04062-f002:**
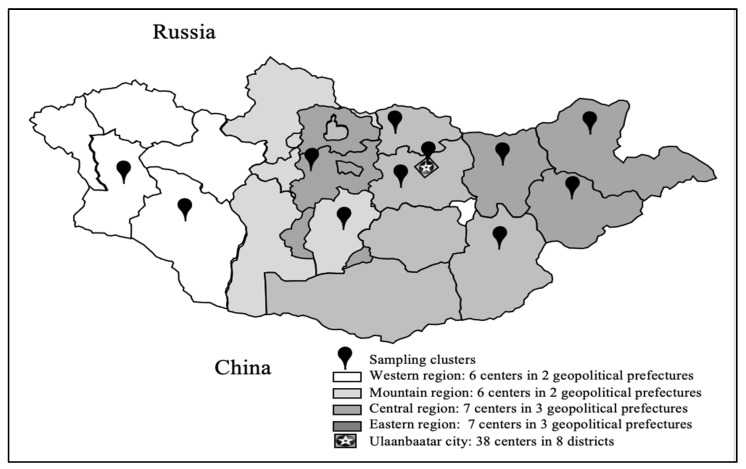
Sampling clusters.

**Figure 3 nutrients-13-04062-f003:**
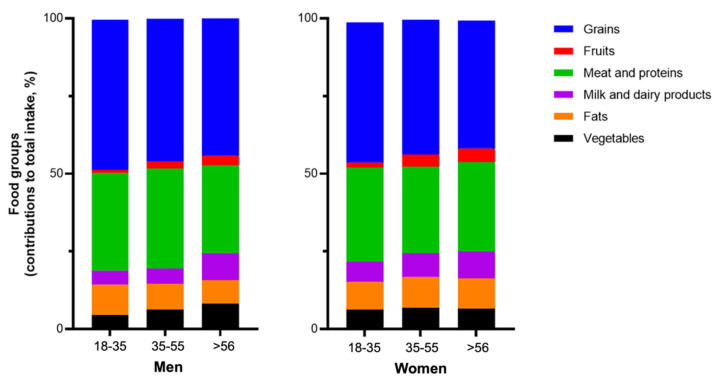
Contribution of food groups to total energy intake by gender and age (years).

**Figure 4 nutrients-13-04062-f004:**
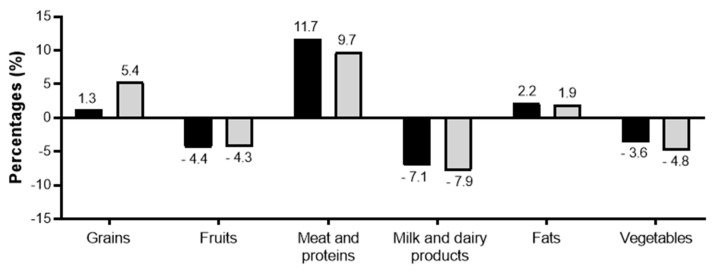
Over and under consumption of food groups regarding the “Ger Recommendations”.

**Table 1 nutrients-13-04062-t001:** General characteristics of study population.

Findings	Total (*n* = 498)	Men (*n* = 108)	Women (*n* = 390)	*p*-Value
Age (year)	43.1 ± 13.1	42.8 ± 14.3	43.2 ± 12.8	0.121
Education: low level, *n* (%)	166 (33.3%)	40 (37.0%)	126 (32.3%)	0.403
Married or cohabitant, *n* (%)	320 (64.2%)	53 (50.9%)	123 (68.4%)	0.175
Living area: Rural *n* (%)	120 (24.9%)	29 (26.8%)	88 (22.6%)	0.148
Family income level: Low, *n* (%)	76 (15.3%)	19 (17.6%)	56 (14.3%)	0.310
Current smokers, *n* (%)	46 (9.2%)	29 (26.6%)	17 (4.3%)	**0.001**
Alcohol use, *n* (%)	107 (21.4%)	39 (36.1%)	67 (17.2%)	0.558
Physically inactive, *n* (%)	144 (28.9%)	35 (32.4%)	109 (27.9%)	**0.010**
Body mass index (kg/m^2^)	27.1 ± 5.1	26.8 ± 5.0	27.2 ± 5.2	0.416
Systolic BP (mm Hg)	124.2 ± 19.0	126.3 ± 16.5	123.3 ± 18.5	0.287

Data are presented as mean ± SD and number (percentages, %). Bold values denote statistical significance at the *p* < 0.05 level.

**Table 2 nutrients-13-04062-t002:** Contribution of food groups to total energy intake by gender and living area.

Food Groups	Contribution to Total Energy Intake (%)								
	Total			Urban			Rural		
	Men	Women	*p*-Value	Men	Women	*p*-Value	Men	Women	*p*-Value
Grains	46.4	43.9	0.064	42.8	38.2	0.096	49.9	47.6 *****	0.481
Fruits	2.0	3.4	**0.010**	2.6	5.6 *****	**0.015**	1.7	2.3	0.498
Meat and proteins	31.0	28.9	0.085	29.9	31.6	0.534	29.3	28.3	0.688
Milk and dairy products	5.8	7.5	**0.049**	6.3	7.1	0.601	6.9	6.9	0.984
Fats	8.6	9.6	0.187	12.1 *****	10.7	0.380	7.1	9.3	0.096
Vegetables	6.1	6.7	0.360	6.2	6.8	0.668	5.1	5.6	0.692

Data are presented as percentages. Bold values denote statistical significance at the *p* < 0.05 level; * indicates the differences between urban and rural participants.

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
