# Peer review of "The Contributions of Food Groups to the Daily Caloric Intake in Mongolian Population: A Mon-Timeline Study"

_nutrients, 2021, doi:10.3390/nu13114062_

Round 1
Reviewer 1 Report
Food group contributions to daily caloric intake among Mongolian population: Montimeline cohort study
I have the following suggestions for improvement:
Title
Line 3 - Mon-TimeLine study? I suggest that the topic should be rephrased in a way that reads better. Something like ‘The contributions of food groups to the daily caloric intake in Mongolian population: A Mon-TimeLine study’.
Abstract
- Rephrase the abstract so it contains your main conclusion and implication to practice. Of what benefit would your article be for the Mongolian population, if they are fortunate to read about your findings? It is good that the authors found out that the ratio of the six macronutrients in the daily caloric intake of Mongolians is unsatisfactory and called for further studies on the subject, but what are the interventions they are recommending for the population and probably for the policymakers?
- Line 12 - write six instead of 6 and do this throughout the manuscript. Numbers 1 to 9 are preferably written in words in articles.
- Line 17-18. Should the summation of the percentages not be equal to 100%?
Introduction:
- Line 29 -31. Any academic reference on significant changes in lifestyle and eating behaviours? You cannot make such a strong statement without any reference!
- Line 60-63, the example you gave is in isolation, and did not flow with the previous statements
- The points in paragraph two (Line 44-68) were not properly and coherently conjoined. The authors did not link ger recommendations/guidelines and the non-communicable diseases accurately.
- There is nowhere you talked about ‘Mon-TimeLine study’ in your Introduction Section
Methods
- Line 77-80, the reason is not scientific, you should have done a sample size calculation! It would have been worthwhile to signpost survey under ‘data source’
- Line 79- … eight aimags or provinces… (seems ‘province’ is more conventional).
- Line 80, it would have been interesting to state a few of the exclusion criteria that drop 498 out of your study.
- How many Figure 1 do you have? Seems one is in the Introduction and another in the Methods section
- Line 84 (supposedly Figure 2) should be properly referenced.
- Was your questionnaire validated? Or how did you test for its reliability?
- What is the justification for 2.3 and 2.4?
Results
- Line 81-82, you said 498 people were 10 excluded from the study based on the exclusion criteria and you later said they were included in Line 150. Not sure which one to go with.
Discussion
- I would not add Line 210 and 215, I could not follow through with what you were saying and how these perfectly fit into what you were presenting.
- Line 222-224, but you only provided a few to them for assessment, how did you come to that conclusion?
- Line 241-246, you made strong statements without any reference. How did you know those assertions?
- I find ‘our country’ very inappropriate. You could refer to your research in that manner, but not the country. For example Line 254 and 294
- You did not clearly show how you arrive at that conclusion in Line 314-319.
Conclusion
- Make the contribution of your findings clearer. Write out very clearly and point by point what you conclude, add an implication section.
- I do not feel your conclusion is giving an answer to this question. Why do you think this study was valuable/original/different from what was done before? This is what you need to address in the conclusion section.
- Line 22, and Line 324 – 330, we already know that the ratio of macronutrients in the daily caloric intake of Mongolians is unsatisfactory, so what is this work bringing into the already existing wealth of information?
- Your conclusion should include what you set to achieve in Line 71-73
Other comments
- I recommend that the manuscript should be proofread by someone with good command of the English language or use Grammarly or other English language correction tools.
- Syntax errors should be corrected by the authors. For example, Line 33-34 and Line 39 (70.[3]) should be properly punctuated.
- Acronyms should be written in full when first used in a write-up. Line 36-38, what is NCD, STIs, CVDs?
- Line 46 ….’In our country, the….’ Which country are you referring to?
- Line 47, 6 should be written as six
At the very least, the abstract, methods and conclusion sections should be revamped!
References
The below reference is odd!
25 AO; UNICEF; UNDP. Joint Food Security Assessment Mission to Mongolia; Food and Agriculture Organization of the United Nations: Ulaanbaatar, Mongolia, 2017.
Author Response
To all editors and reviewers,
Thank you very much for giving us the opportunity to improve and resubmit our manuscript. We sincerely appreciate all constructive and valuable comments and suggestions. Your guidance has helped improve our manuscript.
Sincerely yours,
Oyuntugs Byambasukh
On behalf of co-authors

Reviewer 2 Report
Good work, but the write could be improved. Detail on sampling and selection of participant is needed. The result section should be better organised. Please refer to the suggestions in the attachment.
Author Response

(The authors gave the same response as above.)

Round 2
Reviewer 1 Report
Abstract:
Line 13. Following the “Ger Recommendation”, has been advised.
Comment: This seems to be an incomplete statement
Line 19. (3) Results: The mean age of the study participants was 43.9 ± 12.9 and 21.8% (n = 110) were male.
Comment: The statement would read better as The mean age of the male participants is 43.9 ± 12.9
Line 23-24. It has been observed that the older a person ages, the closer they are to following these recommendations
Introduction
Line 34-35. Mongolia is a landlocked country in Central Asia, located between two great powers, China, and Russia
Comment: Is it necessary to describe the two countries as great powers? Of what relevance is this to your study?
Line 65. Mongolians have been aware of the “Ger recommendations” for at least 20 years
Comment: At least?
Methods
Line 110-111. Finally, a total of 498 people participants were
Line 162-163. Education and family income variables were categorized as low, medium, or high levels., separately.
Comment: I would remove 'separately'.
Results
Line 189 - A total of 498 people participants were included
Discussion
Line 267-268. According to statistics, by 2020, 267 potato cultivation in Mongolia will provide 100% of the country's supply
Comment: Not sure if the authors meant a year in the future? Sounds like a projection
Conclusion
The conclusion section of the manuscript reads like a list of recommendations. I advise that the authors provide the synopsis summary of their findings as a conclusion, and put the recommendations just above the conclusion section.

Reviewer 2 Report
All previous comments are well addressed in the revision.
Author Response
Dear Reviewer,
Thank you very much. Your guidance has helped improve our manuscript. Thank you very much once again for giving us the opportunity to improve and resubmit our manuscript.
Best regards,
Oyuntugs Byambasukh